# Relating graph auto-encoders to linear models

## Abstract

Graph auto-encoders are widely used to construct graph representations in Euclidean vector spaces. However, it has already been pointed out empirically that linear models on many tasks can outperform graph auto-encoders. In our work, we prove that the solution space induced by graph auto-encoders is a subset of the solution space of a linear map. This demonstrates that linear embedding models have at least the representational power of graph auto-encoders based on graph convolutional networks. So why are we still using nonlinear graph auto-encoders? One reason could be that actively restricting the linear solution space might introduce an inductive bias that helps improve learning and generalization. While many researchers believe that the nonlinearity of the encoder is the critical ingredient towards this end, we instead identify the node features of the graph as a more powerful inductive bias. We give theoretical insights by introducing a corresponding bias in a linear model and analyzing the change in the solution space. Our experiments show that the linear encoder can outperform the nonlinear encoder when using feature information.

## 1 Introduction

Many real-world data sets have a natural representation in terms of a graph that illustrates relationships or interactions between entities (nodes) within the data. Extracting and summarizing information from these graphs is the key problem in graph learning, and representation learning is an important pillar in this work. The goal is to represent/encode/embed a graph in a low-dimensional space to apply machine learning algorithms to solve a final task.

In recent years neural networks and other approaches that automate learning have been employed as a powerful alternative to leverage the structure and properties of graphs (Tang et al., 2015; Grover & Leskovec, 2016; Ribeiro et al., 2017; Dong et al., 2017; Hamilton et al., 2017a; Chamberlain et al., 2017; Cao et al., 2016; Wang et al., 2016; Chang et al., 2015). For example, Hamilton et al. (2017b) review key advancements in this area of research, including graph convolutional networks (GCNs). Graph auto-encoders (GAEs), first introduced in Kipf & Welling (2016b), are based on a graph convolutional network (GCN) architecture. They have been heavily used and further refined for representation learning during the past couple of years (see Li et al. (2021); Pan et al. (2018); Vaibhav et al. (2019); Davidson et al. (2018), to name a few). Recently, Salha et al. (2020) discovered that the original GCN architecture, which has been used as a basis for several enhancements, can be outperformed by simple linear models on an exhaustive set of tasks. The equivalence or even superiority of (almost) linear models has also been empirically shown for standard GCN architectures for link prediction and community detection (Wu et al., 2019). The question arises as to why we still use and improve these models when simpler models with comparable performance exist. What is the theoretical relation between linear and nonlinear encoders, and does the empirical work provide a reason to replace graph convolutional networks with linear equivalents? We contribute to understanding the underlying inductive bias of the models. We analyze the influence of the nonlinearity as well as the features on the encoder of the model.

**Contributions:** (1) We draw a theoretical connection between linear and relu encoders and prove that, under mild assumptions, for any function $f(A, X)$ on a graph, there exists an equivalent linear encoder. We use this result to show that the representational power of relu encoders is, at most, as large as the one of linear encoders. (2) We investigate the bias that is relevant for good generalization and give theoretical

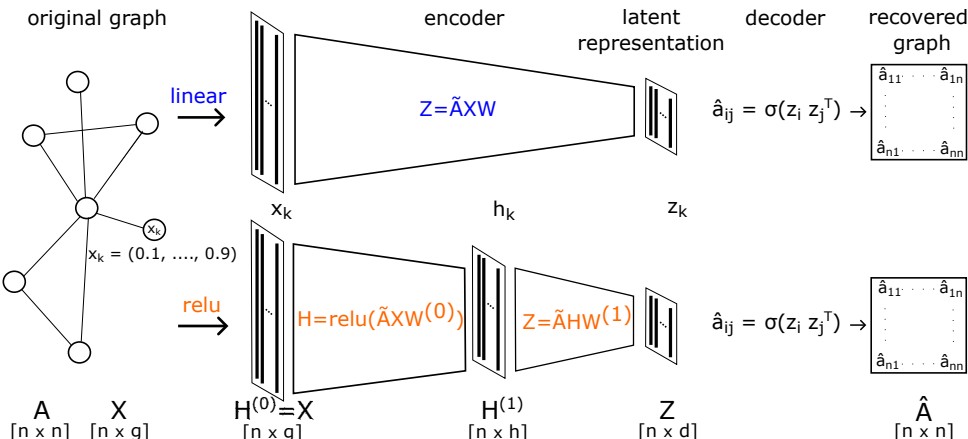

Figure 1: The architecture of a linear and a relu auto-encoder. The only difference is the encoder function: The linear encoder is a simple linear map. The relu encoder is a two-layer GCN network with relu activation in the first layer and linear output activation. In both cases, the input is the graph $(A, X)$. $H^{(1)}$ is the hidden layer of the GCN. A row in the matrix $Z$ gives the latent representation of the respective row/node in the input matrix $X$. The decoder is the inner product, and the target is to recover the adjacency matrix $A$.

insights into how features influence the solution space of a linear model. (3) We empirically find that the nonlinearity in the relu encoder is not the critical ingredient for generalization. Indeed we find that if the graph features are taken into account appropriately, the linear model outperforms the relu model on test data.

## 2 Preliminaries

We consider an unweighted undirected graph $G$ with adjacency matrix $A \in \{0, 1\}^{n \times n}$ and a feature matrix $X \in \mathbb{R}^{n \times g}$. The feature matrix $X$ contains $g$-dimensional feature vectors that describe additional properties of each node of the graph. For example, if the graph consists of a social network, the features could describe additional properties of the individual persons, such as age and income. If we do not have any feature information for the nodes ("featureless graph"), then we set $X$ as the $n \times n$ identity matrix. We always assume that all nodes are connected to themselves, that is, $a_{ii} = 1$ for all $i$. We call $D \in \mathbb{N}^{n \times n}$ the degree matrix with diagonal entries $d_{ii} = \sum_{j=1}^{n} A_{ij}$ and 0 everywhere else, and $\tilde{A} = D^{-1/2} A D^{-1/2}$ the symmetrically normalized adjacency matrix.

**Graph convolutional networks.** Inspired by convolutional neural networks, graph convolutional networks (GCNs Kipf & Welling (2016a)) are one of the most widely used graph neural network architectures. The input consists of a graph $(A, X)$. The graph convolution is applied to a matrix $\tilde{A}$ that encodes the adjacency structure of the graph. This matrix $\tilde{A}$, called diffusion matrix, can, for example, be the (symmetrically normalized) adjacency matrix or the graph Laplacian. In each layer of the network, a node in the graph updates its latent representation, aggregating feature information within its neighborhood, and learns a low-dimensional representation of the graph and its features. More precisely, in layer $l$, each node update is some function of the diffusion of the weighted features: $H^{(l+1)} = \psi(\tilde{A} H^{(l)} W^{(l)})$ with learnable weights $W^{(l)}$, (nonlinear) activation function $\psi$ and $H^{(0)}$ as the feature matrix $X$. The output of the GCN for each node in the graph is a $d$-dimensional vector representing this node's embedding. This representation can be used for downstream tasks such as node, graph classification, or regression.

**Graph auto-encoders.** Graph auto-encoders aim to learn a mapping of the graph input $(A, X)$ to an embedding $Z \in \mathbb{R}^{n \times d}$. This embedding is optimized such that, given a decoder, we can recover the adjacency matrix $A$ from the embedding. As visualized in Figure 1, the encoder consists of a two-layer graph convolutional network with no output activation and hidden layer with size $h \in \mathbb{N}$; the decoder is a simple

dot product:

$$\underbrace{Z_{\text{relu}} = \text{GCN}(X, A) = \tilde{A} \, \text{relu}(\tilde{A}XW^{(0)})W^{(1)}}_{\text{encoder}}, \quad \underbrace{\hat{A}_{\text{relu}} = \sigma(Z_{\text{relu}}Z_{\text{relu}}^{\mathsf{T}})}_{\text{decoder}}, \tag{1}$$

Here $Z_{\text{relu}} \in \mathbb{R}^{n \times d}$ is the latent space representation with embedding dimension $d$, and the matrix $\hat{A}$ is the reconstructed adjacency matrix. Note that $\hat{A}$ is a real-valued matrix that aims to capture the likelihood of each edge in the graph. To obtain a binary adjacency matrix as a final result, we discretize by thresholding the values at 0.5. The loss function optimized by the auto-encoder is the cross-entropy between the target matrix $A$ and the reconstructed adjacency matrix $\hat{A}$. It is minimized over the encoder's parameters $W^{(l)}$. The decoder is a deterministic function and is not learned from the data. For a graph with adjacency matrix $A$, latent space embedding $Z$, and with $\sigma(\cdot)$ denoting the sigmoid function, the loss is formalized as follows:

$$l_A(Z) = \sum_{ij=1}^{n} a_{ij} \log(\sigma(z_i z_j^{\mathsf{T}})) + (1 - a_{ij})(1 - \log(\sigma(z_i z_j^{\mathsf{T}}))) \tag{2}$$

Below we compare GAEs based on a relu encoder with a simplified model based on a linear encoder.

## 3 Relu encoders have at most the representational power of linear encoders

Graph auto-encoders are widely used for representation learning of graphs. However, it has also been observed that linear models outperform them on several tasks. In this section, we investigate this empirical observation from a theoretical point of view: We define a graph auto-encoder that uses a simple linear map as the encoder and prove that the derived model has larger representational power than a relu encoder. To this end, we introduce a linear encoder based on a linear map of the diffusion matrix $Z_{\text{lin}} = \tilde{A}W$, where $Z_{\text{lin}}$ is the latent space representation. The loss function and the decoder remain as in the relu model. For a relu encoder, the dimension $h$ of the hidden layer then satisfies $g > h > d$ (for the linear encoder, the parameter $h$ does not exist). We now define the solution spaces to help us describe the respective models' behavior. They represent the embeddings that can be learned by relu and linear encoders, both with and without node features:

$$\mathcal{Z}_{\text{lin}} = \{Z \in \mathbb{R}^{n \times d} | Z = \tilde{A}W \text{ for } W \in \mathbb{R}^{n \times d}\}$$
$$\mathcal{Z}_{\text{lin},X} = \{Z \in \mathbb{R}^{n \times d} | Z = \tilde{A}XW \text{ for } W \in \mathbb{R}^{g \times d}\}$$
$$\mathcal{Z}_{\text{relu}} = \{Z \in \mathbb{R}^{n \times d} | Z = \tilde{A} \, \text{relu}(\tilde{A}W^{(0)})W^{(1)} \text{ for } W^{(0)} \in \mathbb{R}^{n \times h}, W^{(1)} \in \mathbb{R}^{h \times d}\}$$
$$\mathcal{Z}_{\text{relu},X} = \{Z \in \mathbb{R}^{n \times d} | Z = \tilde{A} \, \text{relu}(\tilde{A}XW^{(0)})W^{(1)} \text{ for } W^{(0)} \in \mathbb{R}^{g \times h}, W^{(1)} \in \mathbb{R}^{h \times d}\}$$

It is easy to see that $\mathcal{Z}_{\text{lin},X} \subseteq \mathcal{Z}_{\text{lin}}$. In words: when we compare two linear encoders, one with node features and one without node features, the one with node features has a more restricted solution space. The reason is that the feature matrix $X$ is at most rank $g$ and typically low rank compared to the diffusion matrix $\tilde{A}$, which restricts the image of $\tilde{A}X$. The same holds for the relu encoder: $\mathcal{Z}_{\text{relu},X} \subseteq \mathcal{Z}_{\text{relu}}$. The relation between the linear encoder and the relu encoder is less obvious. One might guess that the relu encoder can learn more mappings than a linear encoder. However, we now show that this is not the case: we will see that $\mathcal{Z}_{\text{lin}} \supseteq \mathcal{Z}_{\text{relu}}$, focusing on the featureless model. This relation implies that the linear encoder has a larger representational power than a relu encoder.

This paper focuses on a specific architecture introduced in previous work and heavily used in practice. However, under mild assumptions on the input graph, our insights extend to any encoder function $f$. Thus, we state our result in a more general way. We consider the typical case where the number of nodes $n$ is larger than the feature dimension $g$, and we assume the goal is to map each input to a low dimensional space $d$, which means $n > g > d$.

**Theorem 1 (Solution space).** *Given a fixed graph $(A, X)$. We describe a node $i$ in the graph by its neighborhood structure $a_i$ and its features $x_i$; here $a_i, x_i$ are the rows in the corresponding matrices $A$ and $X$. Let $f : \mathbb{R}^n \times \mathbb{R}^g \to \mathbb{R}^d$ be an arbitrary function that maps a point $(a_i, x_i)$ to a low dimensional embedding. Let $g_{\tilde{W}} : \mathbb{R}^n \times \mathbb{R}^g \to \mathbb{R}^d, (a_i, x_i) \mapsto a_i \tilde{W}$ be a linear map parameterized by $\tilde{W} \in \mathbb{R}^{n \times d}$ that maps the neighborhood information $a$ of a point $(a, x)$ to a low dimensional embedding. If $A$ is full rank, then for any function $f$, there exists a matrix $\tilde{W}$ with $f(a_i, x_i) = a\tilde{W} = g_{\tilde{W}}(a_i, x_i)$ for all nodes $(a_i, x_i)$ in the graph. That is, for any function $f$ there exists an equivalent linear map $g_{\tilde{W}}(a_i, x_i)$.*

*Proof.* For simplicity, we consider the functions applied to the whole train set at once:

$$f^\square : \mathbb{R}^{n \times n} \times \mathbb{R}^{n \times g} \to \mathbb{R}^{n \times d}, f^\square(A, X)_{:i} := f(a_i, x_i) \text{ and } g^\square_{\tilde{W}} : \mathbb{R}^{n \times n} \times \mathbb{R}^{n \times g} \to \mathbb{R}^{n \times d}, g^\square_{\tilde{W}}(A, X)_{:i} := g_{\tilde{W}}(a_i, x_i).$$

Let $W_A = A^{-1} f^\square(A, X) \in \mathbb{R}^{n \times d}$. We chose $\tilde{W} := W_A$ to derive that $f^\square(A, X) = A A^{-1} f^\square(A, X) = A W_A = g^\square_{W_A}(A, X)$. It follows that $f(a_i, x_i) = g_{W_A}(a_i, x_i)$ for all nodes $(a_i, x_i) = (A, X)_{:i}$ in the graph which concludes the proof. $\square$

The graph auto-encoder architecture is one particular case of $f(a, x)$:

**Proposition 2 (Representational power of GAEs).** *For any (trained) graph auto-encoder in $\mathcal{Z}_{\mathrm{relu}} \supseteq \mathcal{Z}_{\mathrm{relu},X}$, there exists an equivalent, featureless linear encoder in $\mathcal{Z}_{\mathrm{lin}}$, that can achieve the same training loss: $\mathcal{Z}_{\mathrm{lin}} \supseteq \mathcal{Z}_{\mathrm{relu}}$.*

*Proof.* Let $Z_{\mathrm{lin}}(W) = \tilde{A} \cdot W$ for $W \in \mathbb{R}^{n \times f}$ be the latent representation of the linear model with weights $W$. We can write the latent representation for any GAE as a linear function of the form $Z_{\mathrm{relu}}(W) = \tilde{A} \cdot W$ for some matrix $W \in \mathcal{W}$, where $\mathcal{W}$ is the set of matrices that can be represented by the relu term in the function: $\mathcal{W} = \{\mathrm{relu}(\tilde{A} X W^{(0)}) W^{(1)} : W^{(0)} \in \mathbb{R}^{n \times d}, W^{(1)} \in \mathbb{R}^{d \times f}\} \subseteq \mathbb{R}^{n \times f}$. We now define the possible latent embeddings that the two models can learn. For the relu encoder we have $\mathcal{Z}_{\mathrm{relu}} = \{Z_{\mathrm{relu}}(W) \text{ for all } W \in \mathcal{W}\}$, and for the linear model we get $\mathcal{Z}_{\mathrm{lin}} = \{Z_{\mathrm{lin}}(W) \text{ for all } W \in \mathbb{R}^{n \times g}\}$. Since $\mathcal{W} \subseteq \mathbb{R}^{n \times g}$, the set of learnable embeddings of the relu encoder is a true subspace of all learnable embeddings of the linear encoder: $\mathcal{Z}_{\mathrm{relu}} = \{\tilde{A} W : W \in \mathcal{W}\} \subseteq \{\tilde{A} W : W \in \mathbb{R}^{n \times g}\} = \mathcal{Z}_{\mathrm{lin}}$. It follows that, given any loss $l$, the linear model can achieve at least as good performance as the relu encoder: $\inf_{Z_{\mathrm{relu}} \in \mathcal{Z}_{\mathrm{relu}}} l(Z_{\mathrm{relu}}) \geqslant \inf_{Z_{\mathrm{lin}} \in \mathcal{Z}_{\mathrm{lin}}} l(Z_{\mathrm{lin}})$. $\square$

This proposition has a simple but strong implication: the featureless linear model is, in principle, more powerful than the traditional GAE (with or without features). Here "more powerful" means that the linear model can achieve better training loss.

**Consequences.** To better understand the implications of Proposition 1, it is instructive to engage in the following thought experiment. Consider the standard supervised learning setting with a training data matrix $U \in \mathbb{R}^{n \times d}$ and labels $y \in R$. For any function (say a deep neural network) $f : \mathbb{R}^{n \times d} \to \mathbb{R}^n$, it is possible to express $f(U) = U U^+ f(U) = U W$, for some $W \in \mathbb{R}^{d \times 1}$ if and only if $U$ has a right inverse, this is a mild requirement if $d > n$. Note that this is precisely the "underdetermined" or "high-dimensional" setting where it is possible to find a linear map that perfectly fits the training data to the function values. If $d < n$, the right inverse cannot exist; therefore, clearly, one cannot find such a $W$. Let us contrast this with our setting of GNNs in Proposition 2. Since we consider graph networks, the input is the adjacency matrix $\tilde{A} \in \mathbb{R}^{n \times n}$; that is, we have feature dimensions equal to sample size ($n = d$). Our theorem states that when $\tilde{A}$ is full rank, one can similarly find a linear map that perfectly fits the outputs of any nonlinear mapping of the input data. While it may seem surprising at first glance, it is intuitively clear that in high dimensions, one can always find a linear map that perfectly fits the outputs of any nonlinear mapping. As long as the model is restricted to the input data domain, we can attain the same "training loss". With this high-level intuition, we are ready to address the implications of Theorem 1, representing the issue of representational power.

**The issue of representational power.** We do not claim that $\{f : \mathbb{R}^{n \times g} \to \mathbb{R}^{n \times d} \mid f(A, X) \text{ is any nonlinear GNN}\} \subseteq \{f : \mathbb{R}^{n \times g} \to \mathbb{R}^{n \times d} \mid f(A, X) = A W \text{ for } W \in \mathbb{R}^{g \times d}\}$, that is, the function class of linear models contains in the function class of nonlinear models. However, in a setting of a graph neural network, we (usually) have a single graph as an input with a fixed number of data points $n$. In this setting, the goal is to find an embedding of this fixed graph. Our result says $\mathcal{Z}_{\mathtt{relu,X}} \subseteq \mathcal{Z}_{\mathtt{relu}} \subseteq \mathcal{Z}_{\mathtt{lin}}$; the solution space of the nonlinear model is contained in the solution space of the linear one, showing that the representational power of the linear model is larger than the one of the nonlinear model.

**Generalization properties.** Note that Proposition 2 does not say that the model without features outperforms feature-included models. With this proposition, we can not derive anything about the test performance

or the solution that an optimization algorithm might find. So far, we are considering training loss and the existence of a weight matrix $W$. From the above discussion, it is apparent that the linear encoder is less restricted, which means it introduces a weaker inductive bias than the relu encoder. Note that this is independent of the fact that we used the relu activation function or the depth of the convolutional network. The question is whether the restriction that the relu encoder puts on the solution space induces a "good" inductive bias that helps improve the model's test error. As we see below, we empirically find that when using features, a relu encoder does outperform the (featureless) linear encoder on test data. The relu activation and the features both restrict the solution space and introduce an inductive bias. However, which helps to restrict the representational power in a meaningful way?

## 4    The inductive bias of adding features

This section investigates the inductive bias introduced by adding features to the linear model and how they influence the solution space. While Proposition 2 shows the superior representational power of the linear model without features, in Section 5, we will observe improved test performance for models that use feature information. Whenever the features are low rank compared to the diffusion matrix, they restrict the solution space and introduce an inductive bias that facilitates learning and can improve generalization if they hold helpful information. We observe this behavior for both considered architectures, the relu, and the linear encoder. Consequently, we conjecture that the presence of features, and not the relu nonlinearity, introduces the necessary bias to restrict the solution space in a meaningful way. However, some care is needed. As we will see, adding features can also harm performance if they contradict the structure of the target graph. In order to quantify the harm, we derive a measure for misalignment between graph structures and feature information. Based on this, we give theoretical insights about the restriction of the solution space when including features in the linear model.

Intuitively, we call a graph $A$ and the corresponding features $X$ aligned if they encode similar information. In our setting, this would be the case if $A$ is somehow close to $XX^{\mathsf{T}}$: considering the features, $X \in \mathbb{R}^{n \times g}$ as node embeddings, the matrix $XX^{\mathsf{T}}$ describes the reconstructed adjacency matrix, similar as constructed by the dot product decoder. In other words, nodes should be close together in the feature space if and only if an edge connects them. Since the adjacency matrix is symmetric, a decomposition $A = YY^T$ for $Y \in \mathbb{C}^{n \times n}$ always exists. The error that we might introduce by adding features then comes from the fact that $XX^{\mathsf{T}}$ is a (low rank) approximation of $YY^{\mathsf{T}} = A$: Even for perfectly aligned features, the feature dimension $g$ is typically much smaller than $n$. Note that if we consider one graph and two different feature matrices with the same span, both features restrict the solution space in the same way and give rise to the same optimal embedding.

**Proposition 3 (Features with the same span induce the same solution space).** *Let $\tilde{A} \in \mathbb{R}^{n \times n}$ be the diffusion matrix of a fixed graph and $U, F \in \mathbb{R}^{n \times g}$ two different feature matrices for this graph. If* $\mathrm{span}(U) = \mathrm{span}(F)$, *then $\mathcal{Z}_{\mathrm{lin},U} = \mathcal{Z}_{\mathrm{lin},F}$.*

Intuitively, this means that if $U$ and $F$ span the same space, the corresponding models can, in principle, learn the same mappings and output the same embeddings.

*Proof.* We show that w.l.o.g. for every weight matrix $W_U$ with $Z = \tilde{A}UW_U$, there exists a weight matrix $W_F$ with $Z = \tilde{A}FW_F$. From $\mathrm{span}(U) = \mathrm{span}(F)$ it follows that $\mathrm{span}(\tilde{A}U) = \mathrm{span}(\tilde{A}F)$. We define corresponding linear maps $f_U, f_F : \mathbb{R}^d \to \mathbb{R}^n$ with $f_U(x) = \tilde{A}Ux$ and $f_F(x) = \tilde{A}Fx$. Since $\mathrm{Im}(f_U) = \mathrm{Im}(f_F)$ we know that every point $f_U(x) = y$ has at least one pre-image in $f_F$. Let $W_U = [w_{U_1}, ..., w_{U_d}]$ and $Z = [z_1, ..., z_d]$ be such that $z_i = f_U(w_{U_i})$. For every $z_i$ there exists at least one $w_{F_i}$ with $f_F(w_{F_i}) = z_i = f_U(w_{U_i})$. Since this holds for any matrix $W_U$ we can conclude that $\mathcal{Z}_{\mathrm{lin},U} = \mathcal{Z}_{\mathrm{lin},F}$. $\qquad\square$

We have seen that two different feature matrices with the same span restrict the solution space equivalently. Next, we derive a condition for the alignment between features $X$ and graph structure $\tilde{A}$ that quantifies when features are potentially helpful. To do so, we investigate two settings. In the first setting, the features align with the graph structure and can introduce a valuable restriction of the solution space. In the second setting, the features contradict the graph structure and restrict the solution space so that the optimal embedding

can no longer be found. The intuition for helpful features corresponds with the one for good embeddings: If points are connected in the graph, they should have similar feature vectors. Consequently, if we were not to embed the nodes into a low-dimensional space but were simply interested in any latent representation of the graph, the features themselves would represent desirable embeddings.

**Proposition 4 (Features do not hurt if they align with the graph structure).** *Let $X$ be the feature matrix of the graph with diffusion matrix $\tilde{A}$ and let the embedding dimension coincide with the feature dimension, $d = g$. If $\mathrm{Im}(\tilde{A}X) = \mathrm{Im}(X)$, then there exists a weight matrix $W$ with $X = \tilde{A}XW$.*

This proposition shows that if $\mathrm{Im}(\tilde{A}) = \mathrm{Im}(\tilde{A}X)$, then the model can recover the features.

*Proof.* Since $\tilde{A}X$ and $X$ span the same subspace, we can find a $w_2$ for any $w_1$ with $Xw_1 = \tilde{A}Xw_2$ and $w_1, w_2 \in \mathbb{R}^g$. We interpret every column in a matrix as an independent vector to derive that for any matrix $W_1$ we can find a matrix $W_2$ with $XW_1 = \tilde{A}XW_2$ and $W_1, W_2 \in \mathbb{R}^{g \times g}$. $\qquad\square$

The proposition suggests that if the features encode the same structure as the graph's adjacency matrix, they do not negatively restrict the solution space. Now we prove that if the node relations encoded by their features contradict the adjacency structure of the graph, we can neither recover the features nor the adjacency matrix. In this setting, features restrict the function space in a harmful way, preventing the easiest and trivially optimal embedding.

**Proposition 5 (Features can hurt if they do not align with the graph structure).** *Let $X$ be the graph's feature matrix, and assume that the embedding dimension coincide with the feature dimension, $d = g$. Let $\tilde{A}$ be the diffusion matrix and let $Y \in \mathbb{R}^{n \times d}$ be a matrix such that $\tilde{A} = YY^T$. Assume that the graph can be perfectly represented in $g$ dimensions, that is, there exists a $Z \in \mathbb{R}^{n \times g}$ with $ZZ^{\mathsf{T}} = \tilde{A}$. Let $\mathrm{rank}(Y) = \mathrm{rank}(\tilde{A}) = g$.*

1. *If $\mathrm{Im}(\tilde{A}X) \neq \mathrm{Im}(X)$, then there exists no weight matrix $W$ with $X = \tilde{A}XW$.*

2. *If $\mathrm{Im}(\tilde{A}X) \cap \mathrm{Im}(X)^{\perp} \neq \varnothing$, then there exists no weight matrix $W$ with $Y = \tilde{A}XW$.*

The intuition is that if the adjacency matrix and the node features define two different structures, then neither of the structures can be encoded by the linear model. The alignment of the structures is captured by the condition on the image of $\tilde{A}X$ and $X$.

*Proof. 1.* We prove the contraposition: If there exists a weight matrix $W$ with $X = \tilde{A}XW$ then $\mathrm{Im}(\tilde{A}X) = \mathrm{Im}(X)$. Let $\tilde{A} \in \mathbb{R}^{n \times n}$ and let $X \in \mathbb{R}^{n \times g}$ be full rank. Let $W$ be such that $X = \tilde{A}XW$. Since $X$ is full rank, the rank of $W$ is full and thus $\mathrm{Im}(X) = \mathrm{Im}(\tilde{A}XW) = \mathrm{Im}(\tilde{A}X)$. $\qquad\square$

*Proof. 2.* For $Y = \tilde{A}XW = YY^{\mathsf{T}}XW$ to hold we need $W$ to be the $g \times g$ inverse of $(Y^T X)$. We prove that if $\mathrm{Im}(\tilde{A}X) \cap \mathrm{Im}(X)^{\perp} \neq \varnothing$, then $Y^{\mathsf{T}}X \in \mathbb{R}^{g \times g}$ is of rank smaller $g$ and thus not invertible. We again prove the contraposition: If $X^{\mathsf{T}}Y$ is full rank then $\mathrm{Im}(\tilde{A}X) \cap \mathrm{Im}(X)^{\perp} = \{0\}$. Note that $Y^{\mathsf{T}}X$ is full rank by assumption and thus: $\mathrm{Im}(\tilde{A}X) = \mathrm{Im}(YY^{\mathsf{T}}X) = \mathrm{Im}\,Y$. Generally it holds that $\mathrm{Im}(X)^{\perp} = \ker(X^{\mathsf{T}})$. Let $w \in \mathrm{Im}(Y) \cap \ker(X^{\mathsf{T}})$. Since $w$ is in $\mathrm{Im}(Y)$ there exists a $v \in \mathbb{R}^d$ with $w = Yv$. Since $w$ in $\ker(X^{\mathsf{T}})$, it follows that $X^{\mathsf{T}}Yv = X^{\mathsf{T}}w = 0$. This implies $v = (X^{\mathsf{T}}Y)^{-1}0 = 0$ and thus $w = 0$ and thus $\mathrm{Im}(Y) \cap \ker(X^{\mathsf{T}}) = \{0\}$ $\qquad\square$

Note that the dot product is invariant to orthogonal transformations; thus, for the features, every orthogonal transformation can be absorbed in the weight matrix. Proposition 5 also holds for orthogonally transformed features and embeddings. It also means that we cannot recover any other embedding $\tilde{Y} \neq Y$ that recovers the adjacency structure of the graph. Based on all these insights, we propose a misalignment definition between a graph and its node features. Intuitively, we want to say that $A$ and $X$ are misaligned if $\tilde{A}X$ and $X$ do not span the same subspace.

**Definition 6 (Misalignment of graph and features).** Let $\tilde{A}$ be the symmetrically normalized adjacency matrix and $\tilde{X}$ the normalized feature matrix such that each column has $l2$-norm 1. We measure misalignment

of a graph $A$ and its features $X$ using the distance $d_{\text{algn}} : \mathbb{R}^{n \times n} \times \mathbb{R}^{n \times g} \to \mathbb{R}$, where the arccos function is applied to every entry in the matrix:

$$d_{\text{algn}}(A, X) \mapsto \text{Tr}(\arccos(\tilde{A}\tilde{X}\tilde{X}^{\mathsf{T}})).$$

We got inspired to this measure by the Grassman distance of the two subspaces spanned by $\tilde{A}X$ and $X$; it measures the angles between matching eigenvectors. If this distance is low, the alignment between $A$ and $X$ is high. Note that normalizing the features will not change their span, thus not influence the solution space.

**The generalization error of linear encoders is task dependent.** Learning and generalizing can only work if the learning architecture encodes the correct bias for the given task. We now consider two tasks to demonstrate the negative and positive impact of the bias we introduce when using features.

In recent work, the most common task is **link prediction**. For this task, we consider a graph given by its adjacency matrix $A \in \{0,1\}^{n \times n}$ and the feature matrix $X \in \mathbb{R}^{n \times g}$. We construct a train graph $\bar{A}$ by deleting a set of edges from the graph, which we later, during test time, would like to recover. We train to minimize the loss $l(\bar{A}, X)$ while the test loss is given by $l(A, X)$. In our opinion, link prediction is a somewhat strange task when talking about generalization. We assume that the input graph is incomplete or perturbed. During training, we optimize the GAE for an adjacency matrix that we do not wish to recover during test time (because we also want to discover the omitted edges). However, this cannot be encoded in the loss of the auto-encoder (compare Eq. equation 2). For the encoding to discover the desired embedding, the necessary bias to prevent the model from optimally fitting the training data must be somewhere else in the model. We claim that the features introduce this necessary bias.

We consider a second task, **node prediction**. For this task, we consider a graph given by its adjacency matrix $A \in \{0,1\}^{n \times n}$ and the feature matrix $X \in \mathbb{R}^{n \times g}$. We construct a train graph $\bar{A}, \bar{X}$ by deleting a set of nodes from the graph $A$ and the corresponding features from the feature matrix $X$. We train to optimize $l(\bar{A}, \bar{X})$. We then test on the larger graph $l(A, X)$ with the goal of predicting the omitted nodes. The goal is to learn a mapping from the input graph to the latent space that generalizes to new, unseen data.

We intuitively understand that, dependent on the task, the usefulness of the introduced bias varies. Next, we empirically evaluate the influence of features and compare the performance of the linear and the nonlinear encoder with and without the bias of features.

## 5 Empirical evaluation

In this section, we evaluate the influence of node features and the role of their alignment both for linear and relu encoders. Doing so supports the theory that linear encoders outperform relu encoders.

**Setting.** We consider two models for empirical evaluation. The first is the relu encoder defined in equation 1 based on the implementation of Kipf & Welling (2016b), and the second is the linear encoder. Since the relu encoder has two layers and thus two weight matrices, we realize the weights for the linear model accordingly: $Z_{\text{lin}} = \tilde{A}XW^{(0)}W^{(1)}$. This definition does not contradict the one from above; we can simply choose $W = W^{(0)}W^{(1)}$. Regarding the representational power of the model, this makes no difference. However, from an optimization perspective, it can influence the training (see Saxe et al. (2013)), and we aim to compare the models as fair as possible. We use the same objective function as Kipf & Welling (2016b), weighting the edges to tackle the sparsity of the graph and regularizing by their mean squared norm to prevent the point embeddings from diverging. We train using gradient descent and the Adam optimizer for 200 epochs. As is done in previous work, we measure the performance of the auto-encoders by the area under the receiver operating characteristic curve (AUC). Due to sparsity, the accuracy of the recovered adjacency matrix is not very insightful. We use all present (positive) edges from the graph and uniformly sample the same number of what we call 'negative' edges, that is, edges that are not present in the graph's adjacency matrix. As a result, we only use some of the adjacencies entries for evaluation and get a balanced set of present and not present edges in the graph. To get representative results, we ran every experiment 10 times. In real-world datasets, we randomize over the test set, drawing different edges or nodes each time. For the

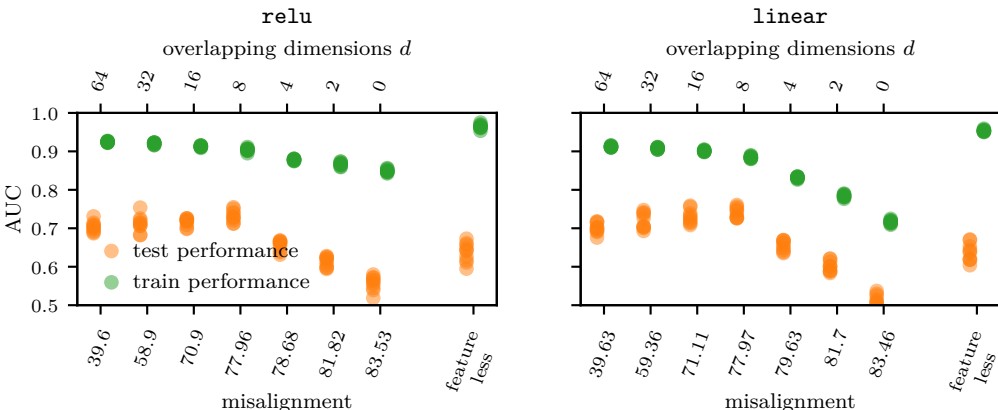

Figure 2: In the link prediction task, features help to regularize if they align with $\tilde{A}$ because they encode the target structure. Along the x-axis we plot the alignment, on the bottom measured by the distance described in Definition 6, low is good alignment, high is bad alignment. On the top we show alignment, given by the number of overlapping dimensions of the two subspaces. The figures show train and test performance for the link prediction task on the synthetic dataset described in Section 5. We plot values for ten independently sampled graphs. We embed the points into eight dimensions.

synthetic datasets, we draw ten independent graphs from the same generative model we describe in 5, then construct the test and train sets on these.

**Datasets.** We use the following generative model to get a **synthetic dataset** with aligned features. We draw $n = 1000$ data points from a standard Gaussian distribution in $g = 64$ dimensions. We normalize the points to lie on the unit sphere and store them in a feature matrix $X \in \mathbb{R}^{n \times g}$. We then construct the graph by thresholding the matrix $XX^\mathsf{T}$ to get a 0-1-adjacency matrix of density between 0.01 - 0.02. With this construction, the feature matrix and the graph adjacency matrix are as aligned as they can be, given the dimensional restriction. We generate data that simulates misalignment as follows. Our goal is to construct features that overlap in $d$ dimensions with the graph's optimally aligned, true features. Recall that changing the span does not restrict the solution space, so we can replace the features with any other basis of the same space. The feature matrix $X$ has $\mathrm{span}(u_1, ..., u_g)$. We construct a perturbed feature matrix $X_{\texttt{perturbed}}$ that has the same rank $g$ and spans the first $d$ dimensions of the space; $\mathrm{span}(u_1, ..., u_d)$, but is orthogonal to the remaining space; $\mathrm{span}(u_{d+1}, ..., u_g)$. We do so by calculating the singular value decomposition $X = U\Sigma V^\mathsf{T}$, then choose the first $d$ columns and the last $g - d$ columns from $U$. Let $U = [u_1, ..., u_n]$, we set $X_{perturbed} = [u_1, ..., u_d, u_{n-(g-d)}, ..., u_n]$. Note that the vectors $u_{n-(g-d)}, ..., u_n$ lie completely in the orthogonal complement of $\mathrm{span}(u_1, ..., u_g)$, which means that the span of the optimal features $X$ and the perturbed features $X_{\texttt{perturbed}}$ overlap in exactly $d$ dimensions. We calculate the misalignment between the graph $A$ and the perturbed features $X_{\texttt{perturbed}}$ as in Definition equation 6.

As **real world datasets** we consider three standard benchmarks: `Cora` , `Citeseer` and `Pubmed`. Similar to previous work, we embed the nodes into 16 dimensions when considering the link prediction task. For the node prediction task, embedding into 16 dimension turns out to be too simple a task. We thus use 4 as the embedding dimension. For graph statistics of all used datasets and more details on the experimental setup see Appendix A.

**When features help for link prediction.** In Figure 2, we plot the AUC performance of both encoder variants, relu and linear, both with and without features, on the link prediction task. We observe that both models' test performances usually decrease with increasing misalignment between the features and the graph. Adding features decreases the train performance, indicating that features restrict the solution space. Features may encode similarities in the feature space that are not present in the (incomplete) graph. This information restricts the solution space and can help to recover those missing structures. Moreover,

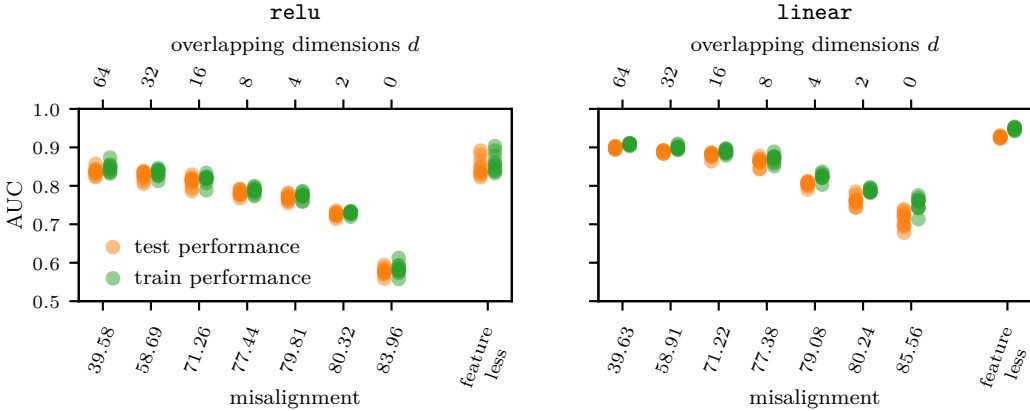

Figure 3: Adding features can harm the node prediction task performance, preventing optimal graph fitting. The x-axis shows the alignment on the bottom measured by the distance described in Definition equation 6; low is good alignment, high is bad alignment. On the top we show alignment given by the number of over-lapping dimensions of the two subspaces. The figures show the performance of training and test performance for the node prediction task on the synthetic dataset described in Section 5. We plot values for ten independently sampled graphs. We embed the nodes into eight dimensions.

the featureless model shows high train performance while the test performance is low. This behavior is expected in link prediction as optimal train performance implies sub-optimal test performance. In Figure 4, we observe the same behavior for the real-world datasets. Adding the features adds additional information about the target structure and improves the test performance. We suggest the following perspective: In the link prediction task, features act as a regularizer preventing the model from encoding the adjacency matrix optimally. If they encode the desired graph structure, this regularization makes the model more robust to perturbations in the adjacency matrix, which is basically the task for link prediction. Here we assume the input graph is incomplete or perturbed, and we want to be robust against these perturbations and still recover the original graph. If the features do not encode the desired structure, including them can harm the performance. However, the relu encoder, where the features are passed through a weight multiplication and a relu activation, turns out to be more robust to false information, possibly learning to ignore (parts) of the features. The linear model can not compensate for erroneous feature information, which becomes visible in Figure 2, where, with more significant misalignment, the training error for the relu encoder drops slowly compared to the linear encoder. We can suspect slight overfitting in both settings as soon as the overlapping dimension exceeds the embedding dimension. The effect is declining test performance, even for very aligned features. In this case, the introduced bias via the features might be too weak.

**When features harm.** In Figure 3, we visualize the performance of the models in the node prediction task. We observe similar behavior for the architectures using features as in the previous task. As expected, for an increasing distance of alignment, the performance decreases. Interestingly, the test performance stays close to the train performance, which indicates that both models generalize well to unseen nodes and larger graphs. Supporting our intuition, we observe that for both encoders, the featureless versions outperform the architectures using features. In the node prediction task, restricting the representational power by adding features harms the model's performance. Our experiments for the node prediction task show that featureless models can outperform the versions including features. Contrary to the link prediction task, in the node prediction setting, we assume that the given graph adjacency matrix already encodes the target structure of every node in the graph. Even if the features align optimally with the adjacency matrix, they do not hold additional information not already encoded in the adjacency matrix. In this setting, features still add a bias, but assuming the given adjacency matrix is noiseless, restrict the solution space in an undesired way.

In practice and real-world data, we usually do not know wether the given features encode the correct structure or if the graph is complete. So not all of the intuition discussed above directly applies. For

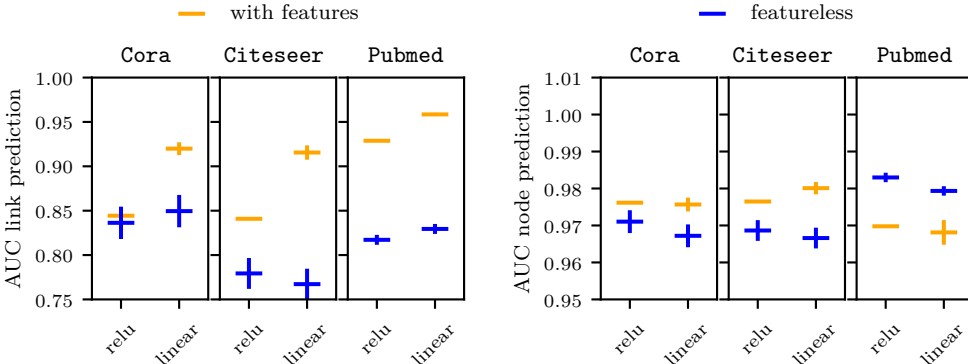

Figure 4: Test performance for the three real-world datasets `Cora Citeseer` and `Pubmed`. Vertical lines indicate standard deviation. We consider relu and linear encoder models with and without features. Left: Link prediction task; we embed points into 16 dimensions. Right: Node prediction task, points are embedded into four dimensions. See Appendix B for the performance when embedding into 16 dimensions.

the three benchmarks and the node prediction task, embedding into 16 dimensions already achieves nearly perfect recovery for all models (not shown in the figures). We embed the data into only four dimensions to get more insightful results. In Figure 4, we see that for the `Cora` and the `Citeseer` datasets, the models using features perform slightly better than those without features. This behavior could indicate noise in the input adjacency matrix, which the feature information regularizes. However, this difference is minimal. Compared to the synthetic dataset and `Pubmed`, the two networks from `Cora` and `Citeseer` have significantly higher feature dimensions in relation to the number of nodes. In summary, our findings indicate that the features encode the correct structure, and adding them can improve optimization even for features with almost no additional information.

## 6   Conclusion

This work presents a theoretical perspective on the representational power and the inductive bias of graph auto-encoders. We consider a relu architecture and a linear architecture for the encoder. We prove that the linear encoder has greater representational power than nonlinear encoders. Theorem 1 also extends to more advanced models with similar encoder architecture and even to other nonlinear graph functions. Based on our experiments and empirical work in the literature, the nonlinear structure of the relu auto-encoder does not improve learning compared to the linear encoder. Our evaluations support the idea that the introduced bias from the nonlinearity is not the crucial ingredient to reducing the solution space in a meaningful way, On the other hand, the features can introduce a powerful inductive bias to both encoder architectures, improving their test performance. Whether features improve training and test performance heavily depends on the task we want to solve. For the two example tasks we consider in this paper, features help with link prediction but do not help node prediction. However, supporting the idea that linear encoders have larger representational power in training and can generalize, the linear encoder outperforms the nonlinear one in both tasks.

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

## A  Experimental details

For our empirical evaluation we consider one synthetic dataset and three real world citation networks; `Cora`, `Citeseer` and `Pubmed`. Table 1 shows the graph statistics for the considered datasets.

| dataset | nodes | edges | density | dim. features |
|---|---|---|---|---|
| `Cora` | 2708 | 5278 | 0.00143 | 1433 |
| `Citeseer` | 3327 | 4614 | 0.00083 | 3703 |
| `Pubmed` | 19717 | 44324 | 0.00023 | 500 |
| Synthetic | $\sim 1000$ | $\sim 10000 - 20000$ | $\sim 0.01 - 0.02$ | 64 |

Table 1: Graph statistics of the real world datasets and averaged values in the synthetic setting.

For every dataset we compute the performances for two different tasks; node and link prediction, and four different models: using the relu encoder and the linear encoder once with features and also without features. So we compute performances in eight different settings for each dataset.

Every graph is split into train, validation and test sets with a ration of 70/10/20. So training always happens on either 70% of the edges and all nodes for the link prediction task, or on only a set of 70% of the nodes for the node prediction task. Test performance is then evaluated on 90% of the edges / nodes. We do not use the parts of the graph used for validation.

The code for the relu graph auto-encoder is publicly available with an MIT license.

We run the experiments on an internal cluster on *Intel XEON CPU E5-2650 v4* and *GeForce GTX 1080 Ti*. All experiments on the synthetic dataset take about 9 hours on single CPU and single GPU. Experiments for `Cora` and `Citeseer` take about 4 and 5 hours respectively. For the `Pubmed`dataset, which is the largest one, running all 8 setups took about 4 days and 4 hours on a single CPU and two GPUs.

## B  Additional experiments

For the real world datasets and the link prediction task we show result for embedding into 4 dimensions in the main paper. Table 2 show results when embedding into 16 dimensions. All models show near to optimal performance which indicates that embedding into 16 dimensions is barely a restriction and too simple of a task.

| cora | | | | |
|---|---|---|---|---|
| | edge | | node | |
| | linear | GAE | linear | GAE |
| features | $0.91 \pm 0.0088$ | $0.88 \pm 0.0123$ | $0.99 \pm 0.0009$ | $0.99 \pm 0.0014$ |
| featureless | $0.84 \pm 0.0078$ | $0.85 \pm 0.0122$ | $0.98 \pm 0.0027$ | $0.98 \pm 0.0016$ |

| citeseer | | | | |
|---|---|---|---|---|
| | edge | | node | |
| | linear | GAE | linear | GAE |
| features | $0.92 \pm 0.0087$ | $0.84 \pm 0.0264$ | $0.99 \pm 0.0010$ | $0.99 \pm 0.0012$ |
| featureless | $0.78 \pm 0.0131$ | $0.78 \pm 0.0149$ | $0.98 \pm 0.0015$ | $0.98 \pm 0.0014$ |

| pubmed | | | | |
|---|---|---|---|---|
| | edge | | node | |
| | linear | GAE | linear | GAE |
| features | $0.96 \pm 0.0018$ | $0.92 \pm 0.0037$ | $0.98 \pm 0.0006$ | $0.97 \pm 0.0055$ |
| featureless | $0.83 \pm 0.0056$ | $0.83 \pm 0.0049$ | $0.99 \pm 0.0009$ | $0.99 \pm 0.0006$ |

Table 2: Test performance in terms of AUC for the three real world datasets `Cora` `Citeseer` and `Pubmed` with given standard deviations. We consider relu and linear encoder models with and without features. Top row: Link prediction task, points are embedded into 16 dimensions. Bottom row: Node prediction task, points are embedded into 16 dimensions.

