# OpenReview forum: "Relating graph auto-encoders to linear models"
_TMLR — Rejected by TMLR_

### Review · Reviewer_VuS1 · 2023-01-14

**Summary Of Contributions:**

The paper conducts a theoretical study on the relationship between linear and non-linear graph auto-encoder (GAE) models, motivated by the observations from the recent literature that, simple linear models can perform on par with the non-linear counterpart. The conclusion is, under certain conditions, the routine ReLU-based non-linear GAE is not more powerful than the linear one. The main approach to reach the conclusion is by checking the solution space between models, in simple words is: checking whether the optimal solution of model A can be ever achieved by model B --> if yes, model A is not more powerful than B.

**Audience:**

Yes

**Claims And Evidence:**

Yes

**Requested Changes:**

Per the above weaknesses, please provide clarification on the full-rank assumption, and evaluation of dense graphs.

**Strengths And Weaknesses:**

Strengths
- The paper provides a useful tool to study the "power" between two models, and an interesting conclusion on the relationship between linear and non-linear GAE models.
- The theoretical study is in-depth: authors analyze the scenarios when node features are absent or not, and when node features align to graph structures.
- Analysis seems to be correct to me, but I might overlook some details.

Weaknesses
- I think the analysis makes a very big assumption without careful justification: "A is full rank". The full-rank assumption greatly enriches the information encoded in the topology, but real-world graphs are always sparse which are hard to be full-rank. I feel this assumption is the foundation of the analysis, thus if it cannot be get around, I would suggest authors first give the clarification and justification, and second find some dense-graph applications for the evaluation -- I think the analysis is still valuable but for a family of graphs rather than generally for all kind of graphs.
- Is the analysis applicable to models with more than two layers?

---

> ### Author Response · Authors · 2023-01-27
> **Initial author response**
>
> Thank you for your time and feedback, we are glad you found this work insightful.
>
>
> - *I think the analysis makes a very big assumption without careful justification: "A is full rank". The full-rank assumption greatly enriches the information encoded in the topology, but real-world graphs are always sparse which are hard to be full-rank.*
>
>     The assumption is mild; see our general comment on the full rank assumption for in-depth discussion.
>
> - *Is the analysis applicable to models with more than two layers?*
>
>     Yes. Observe that Theorem 1 states a quite general result. Given an input graph, we show that an equivalent linear map exists for any function f. This applies to the considered graph auto-encoder with two layers and a relu encoder but does not restrict to this architecture. Theorem 1 applies to any function on a graph that we can represent as a function f(a,x), independent of the number of layers or the activation function. Another example might be a four-layer GCN with a sigmoid activation function.

---

> > ### Comment · Reviewer_VuS1 · 2023-02-10
> > **Thank you for your response, though I am not convinced yet**
> >
> > I have carefully read the response and others' reviews. I am still not convinced by the assumption, and suggest authors should position the analysis for a narrower application rather than overly generic.

---

### Review · Reviewer_uB4X · 2023-01-19

**Summary Of Contributions:**

This paper theoretically studies how the linear and relu encoders are connected with each other in graph auto-encoders. On one hand, the authors prove that a linear encoder has a larger solution space than a nonlinear encoder, which indicates greater representational power. On the other hand, the authors find that the features play an important role in introducing inductive bias on both linear and nonlinear encoder architecture. Empirical evaluations on both synthetic and real-world datasets support the theoretical foundations on both link prediction and node classification tasks.

**Audience:**

Yes

**Claims And Evidence:**

Yes

**Requested Changes:**

My concerns are mainly from the assumption for theoretical proofs, the task gap between theoretical analysis and empirical evaluation and the quantitive results for feature alignment scores on real-world graphs. Please kindly refer to Section Strengths and Weaknesses for detailed suggestions.

**Strengths And Weaknesses:**

Strengths:
* The investigation of the theoretical relationship between linear and nonlinear graph encoders is interesting, which aligns with some practical GNN designs that consecutively remove the nonlinear activation functions [1-2].
* The theoretical foundations proposed in this manuscript are concrete with insights. Experiments are carefully designed and are conducted on both synthetic and real-world graph datasets.
* The paper is generally well-written and almost clear everywhere. I enjoyed reading through this manuscript.

[1] Simplifying Graph Convolutional Networks, ICML 2019

[2] EIGNN: Efficient Infinite-Depth Graph Neural Networks, NeurIPS 2021

Weaknesses:
* The mild assumption that the adjacency matrix $A$ is full rank needs further discussion. It is unclear whether this is the case in real-world situations, ranging from molecules to social networks. Meanwhile, it is also infeasible to check if a social network with massive nodes meets this assumption. I suggest the authors provide more insights into this assumption with perhaps some statistical support.

* Another concern from mine is that the theoretical foundations are proved within the graph generation framework, while the empirical evaluations are performed on link prediction and node classification tasks. This is contradictory in my opinion since the training loss is different and the additional information from the labels might make the analysis more complicated. Could the authors elaborate more on this?

* For the empirical evaluation, the experiments on the synthetic graphs seem more convincing than that on the real-world datasets since the feature alignment scores are not demonstrated. It would be better to add explicit discussions on how the features align with the graph structure in three real-world datasets, which is more informative for employing the theoretical insights into practice.

---

> ### Author Response · Authors · 2023-01-27
> **Initial author response**
>
> Thank you for your detailed feedback! We are glad you enjoyed reading our work.
>
> - *The mild assumption that the adjacency matrix $A$ is full rank needs further discussion.*
>
>     We agree. We refer to our general comment on the full rank assumption for our detailed position on this.
>
> - *Another concern from mine is that the theoretical foundations are proved within the graph generation framework, while the empirical evaluations are performed on link prediction and node classification tasks.*
>
>    Our work focuses on the theoretical analysis of the embedding space and the associated solution space. Indeed, this space does not depend on the loss or the task but solely on the considered model. Consequently, our result holds for any considered loss, and we thus see no contradiction: No matter the downstream task or the loss, if the relu auto-encoder can find a specific solution, so can the linear auto-encoder.
>     On a side note, it is legitimate to question whether the loss captures the correct information for the considered tasks, and we agree that, for example, link prediction is a somewhat weird task given the loss. However, we included this loss and task because the loss is standard for the auto-encoder setting, and link prediction is commonly used for comparison. We discuss our perspective for both tasks in Section 4 of the submitted paper.
>
> - *It would be better to add explicit discussions on how the features align with the graph structure in three real-world datasets, which is more informative for employing the theoretical insights into practice.*
>
>     It is a great suggestion to include the alignment information for the real-world graphs, which we are happy to do. As the alignment depends on the feature dimension, we first normalize by the rank of the features to make the numbers comparable. For the synthetic data sets, you can find the new figures [here](https://noon-aletopelta-65c.notion.site/Relating-graph-auto-encoders-to-linear-models-7a68e8572cb64634874f47ccf88cbc89).
>
>     The normalized misalignment scores for real-world datasets are 0.6709 for cora, 0.8577 for pubmed, and 0.6940 for cseer. Comparing these numbers to the ones of the synthetic datasets, we observe that this range corresponds to almost perfectly aligned features. This supports our hypothesis that the considered graphs show good alignment between the graph structure and the feature information.

---

> > ### Comment · Reviewer_uB4X · 2023-02-09
> > **Thank you for the response**
> >
> > Thank you for the detailed and thoughtful feedback. I have also read the comments from other reviewers as well as the corresponding replies.
> >
> > On the one hand, the additional results on the alignment information for the real-world graphs are interesting and helpful for practical deployment. Hopefully, the authors can include these additional results in the revision. The additional explanations of why link prediction and node classification tasks are selected for empirical evaluation are reasonable, although it would be better to include direct tasks on graphs involving an encoder-decoder style framework such as molecule graph generation.
> >
> > On the other hand, the full rank assumption needs further discussion as also pointed out by the other reviewers and in the authors' general response. Meanwhile, for the link prediction task, while we assume that the input graph is incomplete or perturbed, would it affect the assumption of full rank?

---

> > > ### Author Response · Authors · 2023-02-13
> > > **Thank you. We're happy to clarify.**
> > >
> > > Of course, we will include the additional discussion in our revision.
> > > To answer your question about incompleteness and the rank assumption: It has no effect. Natural graphs like citation- or recommendation networks are incomplete; as discussed, they tend to be full rank equally.
> > > In addition, random perturbations are unlikely to decrease the rank of a graph.

---

### Review · Reviewer_zucb · 2023-01-21

**Summary Of Contributions:**

This work analyzes the theoretical relations between linear embedding and graph autoencoders and aims to explain the observation that linear embedding models are as good as graph autoencoders. The work investigates how feature vectors affect a model's performance in link prediction and node classification tasks.

**Audience:**

Yes

**Broader Impact Concerns:**

No concerns about the broader impact.

**Claims And Evidence:**

No

**Requested Changes:**

Proof errors need to be fixed. There seems to be a more complex relationship between linear embedding models and graph encoders. The experiment needs to be redesigned to gain insights into their relationships.

**Strengths And Weaknesses:**

Strengths:

The work is in the correct direction. I agree that linear models have a strong representation power when we only need to learn representations for one graph.

Weaknesses:

1. The correctness of the work is one major issue. Theorem 1 is a core conclusion of the work, but the proof doesn't seem to be correct -- and there is not an easy fix. In particular, the proof uses the inverse of the adjacency matrix, but many adjacency matrices are not invertible [1].

2. The main focus of the paper is the encoding of the adjacency matrix (Eq 2). But the experiment also investigates the effect of node features in the "node prediction" task. I don't see a clear relationship between "node classification" and the loss in Eq 2.

3. The experiment results do not seem to be very supportive of theoretical analysis. The analysis of the experiment results says "We observe that both models’ test performances usually decrease with increasing misalignment between the features and the graph". But the figure shows the performance first goes up and then drops down. The quantitative evidence to support the theoretical findings should be more solid.



[1] Sciriha, Irene. "A characterization of singular graphs." ELA. The Electronic Journal of Linear Algebra [electronic only] 16 (2007): 451-462.

---

> ### Author Response · Authors · 2023-01-27
> **Initial author response**
>
> Thank you for your feedback on our work.
>
>
> - *The correctness of the work is one major issue. [...] In particular, the proof uses the inverse of the adjacency matrix, but many adjacency matrices are not invertible [1].*
>
>     Sorry to disagree; Theorem 1 is correct. In the theorem, we explicitly assume that the matrix is invertible: "If $A$ is full rank, then ...". While one might wonder whether this is a valuable assumption (see the general discussion above), the theorem and its proof are correct.
>
>
> - *The main focus of the paper is the encoding of the adjacency matrix (Eq 2). But the experiment also investigates the effect of node features in the "node prediction" task. I don't see a clear relationship between "node classification" and the loss in Eq 2.*
>
>     As a first remark, we do not investigate "node classification" in our paper. However, we can answer the question for the node and link prediction tasks that we use in our paper.
>     Indeed, this work focuses on the theoretical analysis of the embedding space and the associated solution space. The solution space only depends on the considered model and is independent of the loss or the (downstream) task.
>     The loss we use is standard for the auto-encoder setting and measures how well the decoder can recover the original adjacency from the low-dimensional embedding. This loss is also indirectly related to the two tasks we consider, "link prediction" and "node prediction". We discuss the tasks and the relation to the loss in detail in Section 4 of our paper.
>
> - *The experiment results do not seem to be very supportive of theoretical analysis. The analysis of the experiment results says "We observe that both models’ test performances usually decrease with increasing misalignment between the features and the graph". But the figure shows the performance first goes up and then drops down.*
>
>     There is indeed a minor error in the description of Figure 3. We will correct this mistake. The test loss does increase slightly and only drops after we raise the misalignment such that the features and the graph align in less than eight dimensions. However, the general interpretation of the figure is still correct, and the observed behavior aligns with our intuition: We embed into eight dimensions, so we only start to lose necessary information as we go below that threshold.

---

> > ### Comment · Reviewer_zucb · 2023-02-01
> > **A quick reply**
> >
> > I did miss the ``if'' clause: "If $A$ is full rank", but a good practice of writing theorems is to put all assumptions up front and use the keyword "Assume" or "Suppose".
> >
> > Please check my detailed comments under your general response.

---

> > > ### Author Response · Authors · 2023-02-03
> > > **Author response**
> > >
> > > Our assumption on the rank was stated upfront in our theorem, and the whole field of logic uses statements of the form "If A then B". The exact formulation of a theorem is a matter of taste, and our wording does follow standard mathematical practice.

---

### Author Response · Authors · 2023-01-27
**General comment on the full-rank assumption**

All reviewers are concerned with the full-rank
assumption in Theorem 1. Sorry for not having discussed the assumption -- it is not strong for the two main reasons we will explain below: (1) Adjacency matrices are typically of (nearly) full rank. (2) If not, there are simple workarounds.

### (1) Adjacency matrices typically have full rank:

Although one might suspect that sparse graphs have a low-rank adjacency matrix, this is typically not the case: there is no relationship between the rank of the spectrum and the sparsity of a graph. We can easiest observe this in the simplest of all random graph models, Erdos-Reny graphs: The spectrum of Erdos Renyi random graphs or stochastic blockmodels is well known and is full rank [3], no matter how sparse the graph is (as long as it is connected; otherwise: see below). Some more results are known for the spectrum of particular random graph models (see [1]).

For natural graphs, such theoretic results do not exist. However, experience shows that similar statements tend to be true.
To make this plausible, we computed the eigendecomposition for the graphs we used in our paper's experiments.
Their respective number n of vertices and ranks r are n=r= 2708 for cora, n=3327 and r=3326 for cseer, and n=r= 19717 for pubmed. They all show a typical behavior: their eigenvalue distribution decays towards zero but never attains zero (the small deviation between r and n  is likely due to numerical issues with the smallest eigenvalues, which play no role in our theoretical analysis). For all practical purposes, we can consider them to be of full rank.

Theoretically, one can construct graphs that do not have full rank, see [4] (by reviewer zucb). However, these graphs must be hand-constructed in a particular way; these effects do not occur in practice.



### (2) Even if the adjacency matrix was not full rank, we could easily fix it:

The main reasons for a graph spectrum to be less than full rank are exceptional cases such as k-partite graphs or so-called twin nodes: two vertices with identical connection patterns have two identical columns in the adjacency matrix. If this happens, one would delete the duplicate vertex from the graph and not lose any information for the graph neural network.

Moreover, matrix A in Theorem 1 can be any matrix representing the graph structure. People often choose the adjacency matrix, but it could also be a robustified adjacency matrix $A = adjacency + \lambda I $ for some small $ \lambda $. This matrix is always full rank. People also often use the graph Laplacian (with rank $n-1$ for connected graphs) as input for graph neural networks, which is also an appropriate choice to represent the neighborhood structure.

[1] On the singularity probability of random Bernoulli matrices, 2008

[2] Nullity of graphs: An updated survey, 2011

[3] The Rank of Random Graphs, 2008

[4] A characterization of singular graphs. 2007

---

> ### Comment · Reviewer_zucb · 2023-02-01
> **The two points doesn't seem to be convincing**
>
> 1. Ranks of random graphs.
>
> I don't understand the statement " The spectrum of Erdos Renyi random graphs or stochastic blockmodels is well known and is full rank". What do you mean by "the spectrum of Erdos Renyi random graphs"? The reference [3] has a clear condition in the abstract about how likely a random graph has full rank. It does need some conditions.
>
> Furthermore, the paper treats each adjacency matrix as given, so I feel that the focus should still be a single matrix, not a distribution. In your investigation, you have three networks, one of which does not have full rank. Can I say one-third of real applications cannot count on your analysis? I feel your analysis certainly has some implications for these applications, but a formal treatment of these cases is needed.
>
> 2. A fix for adjacency matrices without full ranks.
>
> I don't think this statement "... adjacency matrix $A = adjacency + \lambda I$ for some small $\lambda$. This matrix is always full rank" is correct. For a simple graph, the adjacency matrix (denoted by $\hat{A}$) must have eigen vectors corresponding to both negative and positive eigen values.  This is because the sum of all eigen values is the trace of the adjacency matrix and is zero.
>
> Suppose two eigen vectors $\mathbf{u}$ and $\mathbf{v}$ of $\hat{A}$ correspond to eigen values $\alpha$ and $-\beta$. Here $\alpha$ and $\beta$ are both positive. Then
>
> $$
> \begin{align}
> (a \mathbf{u} + b\mathbf{v})^\top  A (a \mathbf{u} + b\mathbf{v}) &= (a \mathbf{u} + b\mathbf{v})^\top (\hat{A} + \lambda I) (a \mathbf{u} + b\mathbf{v}) \\\\
>  &= (a \mathbf{u} + b\mathbf{v})^\top \hat{A}(a \mathbf{u} + b\mathbf{v}) -  \lambda (a \mathbf{u} + b\mathbf{v})^\top (a \mathbf{u} + b\mathbf{v})  \\\\
> &= a^2 \alpha - b^2 \beta - \lambda a^2 - \lambda b^2.
> \end{align}
> $$
>
> If we let $ a^2 = \frac{(\beta + \lambda) b^2 }{\alpha - \lambda}$, then the equation above is zero, which means that $\hat{A} + \lambda I$ is not a full-rank matrix.
>
> I think it is more reasonable to consider $L + \lambda I$, with $L$ being the Laplacian matrix. However, a fix like this does not seem to be trivial.

---

> > ### Author Response · Authors · 2023-02-03
> > **Author response**
> >
> > 1. Question about the Erdos Renyi random graph.
> >
> > We wrote in our previous answer that the spectrum of its adjacency matrix is full rank "as long as the graph is connected". We omitted the technical details, but here they are: An Erdos Renyi random graph is connected with high probability if its connectivity parameter $p$ satisfies $p > (1 + eps)\ log\ n / n$. And this is exactly the condition that is stated in reference [3], Corollary 1.5.: "Let $c$ be a constant larger than 1. Then for any $c\ ln\ n/n < p < 1/2, G(n, p)$ is almost surely non-singular." One can either state this as an asymptotic result as it is done in [3], which then holds almost surely, or one could also state the analogous result for a fixed $n$, which then holds with high probability.
> >
> > 2. Now, how many natural graphs are full rank?
> >
> > Of course, there cannot be a general statement about this without stating which type of natural graph one considers. In the previous response, we argue that if you encounter natural graphs, they tend to have either full or nearly full rank.  For a random graph that follows a specific generating model, one can often prove corresponding statements, as in the case of the Erdos Renyi graph. For arbitrary natural graphs, while it is impossible to prove a formal statement, one can verify it in practice, which we did. We simply took the three graphs that were considered in the paper already, and for all of them, its adjacency spectrum is (nearly) full rank.
> >
> > 3. Fix for the adjacency matrices with nearly full rank.
> >
> > Indeed our notation was misleading; sorry for not being more explicit in our previous response.  What we meant to suggest was $A + \lambda I$ where, in the case of the adjacency matrix, $I$ is not the identity matrix but a noise matrix, for example, where each entry in $I$ has been drawn from the standard-normal distribution or the uniform distribution on [0,1]. Additionally, we could symmetrize this matrix if desired.  With any of these constructions, $I$ is full rank with probability one, and the resulting matrix $A + \lambda I$ is full rank almost surely.
> > For the graph Laplacian L, the choice of $L + \lambda I$ with $I$ the identity matrix also works.
> >
> > 4. Additional note.
> >
> > The specific architecture of the graph auto-encoder we investigate in the paper has a final linear layer, thus the analysis of the representational power of the considered model does not rely on the full rank assumption (see Proposition 2). However, with the full rank assumption, we can show a stronger, more general result (Theorem 1). We discuss the implications and impact in Section 3 of our paper.

---

> > > ### Comment · Reviewer_uB4X · 2023-02-09
> > > **A quick question**
> > >
> > > I have a quick question regarding to the rank check on three citation graphs. According to the authors' comments, the respective number n of vertices and ranks r are n=r= 2708 for cora, n=3327 and r=3326 for cseer, and n=r= 19717 for pubmed. I have checked them by myself using the datasets downloaded from pytorch\_geometric and the function provided by numpy (numpy.linalg.matrix_rank). The respective number n of vertices and ranks r are n = 2708, r = 2408 for cora, n=3327, r = 2788 for citeseer. I also checked the rank for adjacency matrix with self-loop $A + I$, which is a general preprocess step for GCNs. The respective number n of vertices and ranks r are n = 2708, r = 2569 for cora, n=3327, r = 2961 for citeseer. The numbers seem to be quite different. Could the authors expand on this further?

---

> > > > ### Author Response · Authors · 2023-02-13
> > > > **Quick Answer**
> > > >
> > > > This difference is an approximation or numerical problem with the numpy rank method as the eigenvalues approach zero. We observe the same behavior. For our results, we compute the eigenvalues (not the rank directly) and observed that none (or only one for citeseer) are equal to 0: *sum(np.linalg.eigh(adj)[0] == 0)*
> > > >
> > > > To answer your question regarding graphs with self-loops, we computed the number of non-zero eigenvalues: $r=n=2078$ for cora,  $r=3324$ for citeseer (4 zero eigenvalues), and $r=n=19717$ for pubmed.

---

### Decision · Action_Editors · 2023-02-17

**Recommendation:** Reject

**Comment:**

The three reviewers unanimously argued toward rejection of the paper. Overall, there were two main issues:

+ The (strong) full-rank assumption, especially when the graph is randomly perturbed as in the experiments. Moreover, the extension towards a rank-deficient adjacency matrix (mentioned in the rebuttal) does not straightforward.

+ The experiments were not very supportive of the theory (see details in the reviewers' responses).

The reviewers think that positioning the analysis for a narrower application rather than overly generic might strengthen the paper, as mentioned also by the authors.

Given that, unfortunately I cannot recommend acceptance, but I encourage the authors to incorporate the reviewers' feedback and re-submit to TMLR.


**Audience:**

The reviewers agree that the submission addresses an interesting problem for the TMLR audience.

**Claims And Evidence:**

Unfortunately none of the reviewers is convinced with the full-rank assumption in the paper, and they also felt that the experiments were not very supportive of the theory. The reviewers also found various issues in the authors' rebuttal and concluded that the submission needs to go through a substantial revision before publication.